# Robotic-Arm-Assisted Total Hip Arthroplasty: A Review of the Workflow, Outcomes and Its Role in Addressing the Challenge of Spinopelvic Imbalance

**DOI:** 10.3390/medicina58111616

**Published:** 2022-11-09

**Authors:** Andrew Ogilvie, Woo Jae Kim, Rhody David Asirvatham, Andreas Fontalis, Pierre Putzeys, Fares S. Haddad

**Affiliations:** 1Department of Trauma and Orthopaedic Surgery, University College Hospital, London NW1 2BU, UK; 2Hôpitaux Robert Schuman, 2540 Luxembourg, Luxembourg

**Keywords:** total hip arthroplasty, robotic-arm assistance, spinopelvic imbalance, spinal pathology, functional component positioning, virtual range of motion, impingement, stiffness

## Abstract

Robotic-arm-assisted total hip arthroplasty (RoTHA) offers the opportunity to improve the implant positioning and restoration of native hip mechanics. The concept of individualised, functional implant positioning and how it relates to spinopelvic imbalance is an important yet rather novel consideration in THA. There is mounting evidence that a significant percentage of dislocations occur within the perceived “safe zones”; hence, in the challenging subset of patients with a stiff spinopelvic construct, it is imperative to employ individualised component positioning based on the patients’ phenotype. Restoring the native centre of rotation, preserving offset, achieving the desired combined anteversion and avoiding leg length inequality are all very important surgeon-controlled variables that have been shown to be associated with postoperative outcomes. The latest version of the software has a feature of virtual range of motion (VROM), which preoperatively identifies potential dynamic causes of impingement that can cause instability. This review presents the workflow of RoTHA, especially focusing on pragmatic solutions to tackle the challenge of spinopelvic imbalance. Furthermore, it presents an overview of the existing evidence concerning RoTHA and touches upon future direction.

## 1. Introduction

Total hip arthroplasty (THA) has been a well-established and successful procedure for over 60 years, with an ever-growing number of primary operations performed on an annual basis. However, changes in the demographics of patients requiring a total hip replacement, as well as limitations in the conventional method, present new challenges that necessitate a modern solution. With great advances in surgical technology, robotic-arm-assisted THA (RoTHA) presents a pragmatic solution to overcome barriers caused by human error through improving implant positioning and restoring normal hip biomechanics.

Robots were first used in surgery in the 1980s, and subsequently implemented into orthopaedic surgery in the early 1990s for THA. Their evolution has yielded a spectrum of functional modes, ranging from fully automated to a semi-active instrument at the beckon of the surgeon’s hand [1,2]. In all cases, the role of the surgeon remains imperative, as the robot delivers the surgeon’s plan. This fits the etymology of robots; robota is the Czech term for ‘forced labour’, first used in the 1920s to describe the programmable, function-performing machines with which we associate the term today [3]. The first generation of robotic surgery was fully active robot systems, where the robot performed the femoral osteotomy and positioned the implant based on the surgeon’s plan. These fully active robotic systems work autonomously with the surgeon overseeing the surgical workflow and are able to activate an emergency stop if deemed necessary. ROBODOC (Curexo Technology Corporation, Fremont, CA, USA), a CT-based system, was the first one utilised in THA. The workflow allowed customisation of the plan, following which the robot would prepare and mill out the proximal femur; however, manual reaming and conventional instrumentation was used for acetabular reaming and implantation. Other fully active systems included CASPAR (Universal Robotic Systems Ortho, Germany) and ACROBOT (The Acrobot Co., Ltd., London, UK).

However, there was an 18% conversion rate to conventional THA (CoTHA), as well as higher rates of dislocation, revision surgery and soft tissue complications compared to manual THA [1,4]. These issues ignited the progression to the second generation widely used today: semi-active robots. This development offers the surgeon more control, as the robotic arm provides the pre-programmed spatial window for the surgeon to precisely execute the procedure with haptic feedback. The most commonly used semi-active robotic system in THA is the MAKO Robotic-arm Interactive Orthopaedic (RIO) system (Stryker, Kalamazoo, MI, USA) [1,5]. This system offers tactile, audio, and visual feedback in addition to utilising haptically confounded boundaries to guide reaming and acetabular cup positioning. Furthermore, the planned femoral osteotomy site can be displayed on the screen, whereas changes in offset leg length changes can be dynamically presented intra-operatively. Finally, the introduction of the latest software introduced a virtual range of motion (vROM) feature, displaying potential impingement and allowing for fine tuning of the components prior to the definitive implantation.

Traditionally, to plan the positioning of the acetabular component in CoTHA, a pre-operative supine anteroposterior (AP) radiograph of the pelvis is taken. However, this does not consider the dynamic relationship between the pelvis and the spine, and it does not address pre-existing spinopelvic pathologies. The importance of addressing these challenges is highlighted by the association between spinal deformity and increased dislocation [6]. RoTHA constitutes a pragmatic solution to addressing the spinopelvic imbalance by assessing individual patient anatomy pre-operatively by computed tomography (CT) scanning, allowing the surgeon to devise a patient-specific plan based on the three-dimensional reconstructions, incorporating spinopelvic measurements from lateral sitting and standing lumbar spine radiographs [7].

The complex relationship between spinopelvic mobility and component positioning in THA presents a challenge for the arthroplasty surgeon. This review presents the workflow of RoTHA, incorporating spinopelvic measurements into pre-operative planning and aiming to achieve functional implant positioning. The workflow presented in this manuscript is pertinent to the Mako system; however, it could be translatable to other robotic systems. Furthermore, our review outlines the existing evidence concerning RoTHA and touches upon the future direction in the field.

## 2. The Spinopelvic Challenge

Since the pelvis is a dynamic unit within the body, spinopelvic motion should be considered when planning the functional implant positioning. The concept of a functional implant position is the combination of the anatomic cup position implanted in the bone and the postural orientation of the pelvis. An accurate functional cup position determined by patient-specific parameters improves hip stability, which is executed with greater precision by RoTHA [8]. Previously, improved component accuracy within the Lewinnek’s safe zone has been thought to reduce the rates of dislocations; however, this view is now being challenged, as dislocations occur despite the acetabular component being within the aforementioned safe zone. A retrospective cohort study of 9784 patients who underwent total hip arthroplasty showed that overall rates of dislocation were relatively low at 2%; however, 58% of patients who dislocated their hip were found to have their acetabular component within Lewinnek’s safe zone [9]. A systematic review by Seagrave et al. also found conflicting evidence within the literature. Eleven of the twenty-eight studies compared dislocation rates with a combined value of anteversion and inclination; seven of the studies revealed more dislocations occurring within Lewinnek’s safe zone compared to those outside this zone. In concordance, Seagrave et al. reported a greater proportion of dislocating hips had cup positioned within Lewinnek’s zone [10]. There are a multitude of factors which affect the stability of total hip arthroplasty, and the notion of patient-specific safe zones is being popularised by taking into consideration the native spinopelvic anatomy for each individual patient, thus recreating patient-specific hip biomechanics [11].

Under normal circumstances of moving into the sitting position from standing, there is a posterior rollback of the pelvis upon the lumbar spine; this is accompanied by a loss of the lumbar curvature. This is to allow for the requisite flexion of the femur which is expected to be 55–70 degrees [12,13,14]. This spinopelvic motion results in a biological opening of the acetabular cup by increasing functional anteversion and inclination [12,15]. Therefore, if the spinopelvic construct is stiff, there is less posterior rollback, necessitating more hip flexion in sitting and more extension in standing [16], which can increase the impingement risk and therefore the dislocation risk [12,16].

The lumbar spine X-rays are evaluated to determine the sacral slope, defined as the angle between the horizontal and the superior end plate of the S1 vertebrae. The difference in the sacral slope between sitting and standing is used as a measure of movement of the pelvis on the lumbar spine as the hip is flexed from standing to sitting position (Figure 1a,b).

According to Stefl et al.’s [17] classification, there are 5 patterns of spinopelvic mobility: normal, “stuck sitting”, “stuck standing” hypermobile normal and hypermobile kyphotic. In general, if there is a change in the sacral slope of >30 degrees, this is considered hypermobile; 11–29 degrees is normal spinopelvic mobility; and if the change is less than 10 degrees, it is stiff, which increases the chances of post-operative dislocation [17].

The stiff spinopelvic group is further divided into two subgroup patterns dependant on the position in which the spinopelvic joint is fixed: “stuck-standing” if the pelvis is stuck in the anterior position of standing defined by an absolute pelvic tilt of <30 degrees in the sitting position; and “stuck sitting” if pelvis is stuck in the posterior position with an absolute pelvic tilt of <30 degrees when standing because there is no anterior roll of the pelvis normally expected in standing from sitting.

Vigdorchik et al. [18] suggested the hip–spine classification, utilising the pelvic incidence, lumbar lordosis and sacral slope to subdivide patients into four categories [17] (Figure 2).

## 3. Workflow

### 3.1. Pre-Operative Planning and Preparation

“Measure twice cut once” is an old proverb [19], which is now more true than ever with the advances in robotic surgery, especially in orthopaedics and total hip arthroplasty.

The preoperative phase begins with obtaining a CT scan to define the bony landmarks that will later be captured intraoperatively to confirm the anatomy and generate a three-dimensional reconstruction of the femur and pelvis. Additional information regarding the bone morphology in terms of the version of the femur and acetabulum as well as any bony deficiencies can also be collected from the CT. Once the CT data are imported into the robotic software, the robotic product specialist works collectively with the arthroplasty surgeon to create a functional plan. In conjunction with the CT, standing and sitting lateral lumbar spine radiographs are organised to assess spinopelvic motion.

The operative plan is normally finalised in the weeks leading up to the operation; however, adjustments can be made at any time, including during the operation. Last minute checks are usually made by the surgeon in theatres prior to scrubbing for the operation.

The main surgical targets with regards to the acetabular cup positioning that need to be considered are reproducing the native centre of rotation as well as the cup version and size to ensure satisfactory bony coverage (Figure 3a–c). Anatomic anomalies, such as dysplasia, can also be accounted for with the amount of cup expected to be uncovered posteriorly. Bony prominences that could lead to soft tissue irritation or bony impingement are noted. The centre of rotation of the native acetabulum can be reproduced with the implanted cup position with the robotic software, often giving good medio-lateral coverage. The starting cup position in our workflow is 40 degrees inclination and 20 degrees anteversion in line with the Lewinnek and Callanan safe zones [20,21]. The cup size and position can then be altered to ensure good three-dimensional bony coverage and make sure there is no protrusion.

Attention is then turned to the femur as the expected size of the stem, and the planned osteotomy site on the femoral neck can be determined. The robotic software allows careful planning of the femoral head centre of rotation. The estimated effect on changes to the offset and leg length can also be evaluated by comparing to the contralateral and preoperative hip within the robotic software (Figure 4).

The cup and femoral stem positions can be adjusted in conjunction with the spinopelvic motion to ensure no bony impingement is evident, and the combined anteversion is between 25 and 45 degrees to prevent instability [22,23].

The results of the sacral slope calculations from the lumbar spine radiographs are imported into the robotic software, and a virtual range of motion of the joint is performed to evaluate impingement. The virtual range of movement (VROM) tool gives the ability to test the planned hip replacement into positions of maximal range of movement and receive instant feedback in relation to the presence of impingement.

To perform the VROM, the hip is tested in two positions similar to sitting and standing: deep flexion and full extension. In our practice, we test deep flexion at 110 degrees flexion and 40 degrees internal rotation; extension is evaluated at 25 degrees extension and 15 degrees external rotation. In each of these positions, careful evaluation is performed to ascertain the presence of bone-on-bone, bone-on-implant, or implant-on-implant impingement.

Changes in component positioning are subsequently dictated by the VROM, and examples include lateralising the cup or increasing the femoral offset if there is bone-on-bone impingement; increasing the femoral stem version if there is posterior impingement; or planning to remove osteophytes if they are deemed to be the source of impingement.

### 3.2. Intra-Operative Workflow

Following the surgical approach, a screw for the removable femoral array is inserted into the greater trochanter away from the planned femoral osteotomy site. A check point is also placed next to the femoral array screw to confirm the system accuracy immediately before using the robotic arm to ensure that nothing has come loose during the operation.

Bony landmarks are captured on the femur before dislocation on both trochanters, femoral neck, and femoral head. After registration, the femoral osteotomy can be planned using the probe on the femoral neck and marked with diathermy. After the femur is prepared, the femoral broach position can be checked by using ‘Broach Tracking’, which has three pre-registered divots on the trial neck. This allows the surgeon to compare the version, depth, and lateralisation of the broach to the preoperative plan.

Attention is then turned to the acetabulum as landmarks are registered along the edge of the acetabulum rim and inside. A second checkpoint is also placed superior to the acetabulum with a tied suture for retrieval at the end of the operation.

The robotic arm is used to prepare the acetabulum with attached hemispherical reamers. It is locked into position using stereotactic boundaries and is only able to ream in the plane of the planned cup version and inclination. The robotic software provides real-time feedback of the depth of the reaming. The definitive cup is then positioned within the acetabulum in the appropriate version and inclination, and the robotic arm is again locked within the stereotactic boundaries and impacted with a manual mallet. The robotic software displays the depth remaining until the cup is seated. After implantation, the cup position can be checked against the preoperative plan using three pre-registered divots on the definitive cup.

Real-time feedback on the leg length and combined offset changes during the operation are obtained by performing a trial reduction (Figure 5). In this position, the VROM is also tested again to ensure no in vivo impingement. Changes to the implants can be made according to any source in impingement found on the robotic software or any clinical instability that may be evident. Combined anteversion is also displayed and taken into consideration.

### 3.3. Case Presentation

The figures below depict pre-operative planning in a 68-year-old female patient with stiff spinopelvic construct and the changes that had to be made to overcome the challenge of spinopelvic imbalance.
The patient had previously undergone a left total hip replacement complicated by two post-operative dislocations. She did not report a history of back pain.The difference in sacral slope between standing and sitting radiographs was noted to be 6 degrees. According to the Stefl classification, this is stuck sitting, as the sacral tilt does not tilt anteriorly beyond 30 degrees with standing, indicating a high-risk patient (Figure 6a–c).In this case, the native femoral retroversion (−6 degrees) posed a challenge in avoiding impingement (Figure 7). Upon assessing VROM, bone-on-bone and implant-on-implant impingement in deep flexion were noted (Figure 8). Using the robotic software, the planned femoral version was corrected to +16 in the femoral broach (Figure 9a,b).VROM was performed again, and impingement in flexion was eliminated (Figure 10). In extension, upon subtracting the femur, it became apparent that there was a small area of impingement secondary to an anterior osteophyte which was planned to be removed after cup insertion during the operation (Figure 11a,b).The robotic software also enables preoperative and intraoperative visualisation of the anticipated postoperative X-rays accounting for any changes to the plan. In addition, the software allows for calculation of changes to the leg length offset compared to the preoperative and contralateral hips. In this case, the leg length was 1 mm longer compared to the opposite hip, and the combined offset was 6 mm increased compared to that preoperatively (Figure 12a,b).

## 4. Outcomes

### 4.1. Component Accuracy

Historical targets mandated positioning of the acetabular component into 15 ± 10 degrees of anteversion and 40 ± 10 degrees of inclination as described by Lewinnek [20], which has been thought to reduce dislocation rates, component wear and instability. In CoTHA, surgeons utilise anatomical landmarks, such as the transverse acetabular ligament, intra-operatively as a version guide to place the acetabular component [24]. On the other hand, RoTHA utilises pre-operative CT imaging to plan the acetabular component positioning and is executed intra-operatively through virtual mapping of the pelvic anatomy to make the specific osseous cuts.

A cohort study of 75 patients by Kayani et al. demonstrated statistically significant improvement in the accuracy of prosthesis placement within the defined safe zones with RoTHA. They found overall component accuracies to be 96% (24/25) and 92% (23/25) within Lewinnek’s and Callanan’s safe zones, respectively, whereas through conventional means, they were 68% (34/50) and 64% (32/50), respectively [25]. Further to this, RoTHA increased the accuracy in achieving the planned combined offset in patients undergoing RoTHA compared to CoTHA methods, thus restoring native hip biomechanics more effectively (24). Similar results were reciprocated by a cohort study by Clement et al. which revealed that 95% and 97.5% of their RoTHA patients achieved inclination and anteversion within Lewinnek’s safe zone, respectively [26], compared to 81.3% and 83.8%, respectively. Recent studies have suggested that superior component accuracy can be achieved with RoTHA, which in turn can conceptually lead to improved implant survivorship and functional outcomes (Table 1).

### 4.2. PROMs

It has been shown that RoTHA can result in more accurate implant positioning and improved restoration of natural hip biomechanics compared to CoTHA [10] and recent evidence suggests that this could translate to improved functional outcomes. A comparative study by Domb et al. indicates that patients receiving RoTHA had improved Harris Hip Score (HHS), Forgotten Joint Score (FJS-12), Veteran RAND-12 Physical (VR-12 Physical) and 12-Item Short Form Health Survey (SF-12) scores at a minimum of 5 years of follow up [27]. Clement et al. in a similar matched cohort study reported that patients in the RoTHA group had a mean Oxford Hip Score (OHS) of 2.5 points greater than CoTHA but no superior Forgotten Hip Score (FHS) [26]. The findings of a meta-analysis focussing on semi-active RoTHA demonstrated a greater improvement in HHS in RoTHA patients in short- to mid-term follow up [5].

Notwithstanding this, there are systematic reviews showing discordant findings. A systematic review and meta-analysis of 14 studies by Han et al. showed no differences in functional outcomes including HHS, Western Ontario and McMaster Universities Osteoarthritis Index (WOMAC) or the Merle D’Aubigne Hip Score [28]. Similar results were echoed by another systematic review and meta-analysis of 1342 patients, of whom 922 patients were involved in studies which compared CoTHA with RoTHA: both fully active and semi-active robots [29]. Samuel et al. performed a systematic review of 18 studies including fully active and semi-active robots (ROBODOC and MAKO). A pooled analysis demonstrated significantly higher WOMAC scores in RoTHA than CoTHA with a mean difference of −3.57 (95% CI −5.62 to −1.52); however, they found no significant difference in HHS, FJS and SF scores [30]. These systematic reviews share similar limitations, which could mask the positive impacts of current RoTHA systems in use. In two of the systematic reviews, active and semi-active robots are lumped together, introducing considerable bias. Additionally, although Samuel et al. discerned fully active and semi-active robots as separate entities, they utilised data from studies of both to draw upon their conclusions, thereby making their conclusions less illustrative of current practice and potential benefits.

### 4.3. Complications

Reported rates of intra-operative and post operative complications for both CoTHA and RoTHA remain low and comparable within the current body of literature in studies encompassing semi-active robots [27,28,30,31,32]. RoTHA involves additional steps and has been suggested to be associated with longer operative times. However, there is strong evidence to suggest that this significantly improves as surgeons progress through the learning curve [27,28,33].

Recent studies have shown reduced rates of dislocation of RoTHA compared with conventional methods. A large retrospective comparative review by Bendich et al. showed that there is a significant reduction in the rate of dislocations within the first year of index surgery requiring revision surgery [34]. Similar findings were echoed by Shaw et al. who reported that RoTHA is associated with significantly lower dislocation rates compared to CoTHA [35]. Furthermore, authors reported that 46% of patients whose hips dislocated after CoTHA went on to require further revision surgery due to recurrent instability, whereas all dislocations following RoTHA were successfully treated conservatively [35]. Systematic reviews by Ng et al. [5] and Samuel et al. [30] reported no statistical difference in rates of dislocation; however, these were pooled studies including first-generation robots, so it is difficult to draw significant conclusions regarding contemporary robotic-assisted surgery. Some studies have reported higher risk of dislocation [25,28,30,32]; however, this is likely associated with the use of fully active systems and has not been replicated in studies with semiactive systems [4,34,35]. Comparable results have also been documented between the two techniques in relation to leg length discrepancy [27,28,32].

### 4.4. Cost Efficacy

RoTHA is associated with substantial cost pertaining to the installation of the robotic device and software, maintenance and sterilisation, additional imaging, such as CT scans, and training of the staff. Notwithstanding this, the costs could potentially be offset if the documented improvement in implant positioning is shown to increase implant longevity and reduce length of stay. Maldonado et al. implemented a stochastic Markov model to compare the cost effectiveness of RoTHA and CoTHA. The authors found that the cumulative cost difference at 5 years for Medicare patients was GBP 945 less in favour of RoTHA and similarly for private healthcare patients, it was GBP 1810 at 5 years [36]. The model also produced more QALY for RoTHA, meaning that it was less costly and more effective than CoTHA [36]. A large retrospective study exhibits similar findings to those above, whereby they found that the mean costs for robotic-arm assistance were GBP 1684 and GBP 1759 less compared to CoTHA at 90 days and 1 year, respectively, with a reduced length of stay of 3.4 days compared to 3.7 days [31]. However, they found readmission rates for RoTHA patients to be higher by 1.2%, which could potentially offset the costs saved initially [31]. On the contrary, a large retrospective study by Kirchner et al. found that the average inpatient hospital cost for RoTHA was GBP 20,046 ± 6165 compared to GBP 18,258 ± 6147 for CoTHA patients, despite a shorter length of stay: 2.69 +/− 1.25 days in comparison to 2.82 +/− 1.18 days [37]. There is yet to be substantive evidence to determine the cost effectiveness of RoTHA, owing to the lack of well-designed cost-effectiveness studies on this novel procedure.
medicina-58-01616-t001_Table 1Table 1Table of characteristics and results of the studies reporting outcomes comparing conventional and robotic total hip arthroplasty.Author and YearMaterial and MethodsResearch TypeMeasured OutcomesKey ResultsHan et al. [28]2019From Embase, PubMed, Cochrane Library14 studies included: 12 high quality and 2 medium qualitySystematic review and meta-analysisComparing functional outcomes, radiological outcomes and complication rates between CoTHA and RoTHACoTHA had less case of dislocations compared to RoTHA.WOMAC, HHS and Merle D’Aubigne Hip Score showed no statistical difference.Robotic THA resulted in greater number of implants in Lewinnek’s safe zoneDomb et al. [27]2020From total of 217 patients, 66 patients matched into each cohort (RoTHA and CoTHA).Propensity Score-Matched studyComparing PROMs, acetabular implant placement, survivorship and complications between each cohort.RoTHA resulted in improved PROMs (HHS FJS-12, VR-12 Physical, SR-12) compared to CoTHA.Improved implant accuracy within Lewinnek with RoTHA vs. CoTHA (97% and 73.8% respectively)Clement et al. [26]202040 RoTHA patients and 80 CoTHA patients, performed by single surgeonPropensity Score-Matched studyComparing PROMs, implant positioning, patient satisfaction and restoration of leg lengthStatistically significant improvement in OHS score of 2.5 95% CI 0.1–4.8 *p* = 0.038 comparing RoTHA to CoTHA.97.5% of RoTHA components within Lewinnek and Callanan’s safe zoneNg et al. [5]2021Search yielded 510 articles from Medline, PubMed and Google Scholar. 17 includedSystematic review and meta-analysisReport on learning curve, compare implant positioning, survivorship of implants, functional outcomes and complications between semi-active RoTHA and CoTHAImplant accuracy in RoTHA between 77% and 100%, whereas CoTHA between 30% and 82%.Statistically significant improvement in HHS with a mean difference of 3.05 95% CI 0.46 to 5.64Samuel et al. [30]2021Search yielded 526 studies from PubMed, Embase and Cochrane Library. 18 includedSystematic review and meta-analysisComparing PROMs, dislocation, infection and revision rates between RoTHA (MAKO and ROBODOC) and CoTHA.No significant difference in HHS, FJS, SF scores, Merle d’Aubigne but statistically significant improvement in WOMAC MD: −3.57 95% CI −5.62 to −1.52 *p* = 0.006.No difference in revision ratesNo difference in dislocation rates, but statistical difference when comparing ROBODOC to CoTHA.Chen et al. [32]2018Search yielded 178 studies from Medline, Embase, Cochrane Library and other manual sources. 8 included.Systematic review and meta-analysisComparing surgical times, PROMs, complications and radiographic outcomes between RoTHA and CoTHAIntraoperative complications significantly higher in CoTHA than RoTHA with similar post-operative complication rates.No significant difference in PROMs score, surgical times or limb length discrepancies.Improved component accuracy.Karunaratne et al. [29]2019Search yielded 2957 articles from PubMed, Medline, Embase and CENTRALSystematic review and meta-analysisComparing RoTHA and RoTKA PROMs of both fully active and semi-active against conventional.No significant difference in PROMs for both fully active and semi-active RoTHA compared to CoTHA.Kayani et al. [11]201950 CoTHA and 25 RoTHA patients, single surgeonCohort studyComparing accuracy in restoring native centre of rotation, planned combined offset, component accuracy and leg length correction between RoTHA and CoTHA.RoTHA associated with improved accuracy in restoring native hip centres of rotation, improved preservation of native combined offset and acetabular component accuracy.Maldonado et al. [36]2021555 patients who underwent RoTHAUtilised Markov modelCost effectiveness study utilising a Markov modelTo assess the QALY and cost of RoTHA vs. CoTHARoTHA produces more QALY compared to CoTHA (2.96 ± 0.58 and 2.92 ± 0.57)RoTHA Medicare patients was $945 less than CoTHA patients and $1810 less for private insurance patients at 5 years.Kirchner et al. [37]2021758 RoTHA matched against 758 CoTHARetrospective cohort analysisTo assess cost of inpatient careAverage inpatient cost for RoTHA $20,046 ± 6165 compared to CoTHA $18,258 ± 6147.Bendich et al. [34] 202213,802 posterior approach THAs (1770 RoTHA, 3155 computer navigated THA, 8877 CoTHA)Retrospective cohort analysisTo assess rates of complicationLower risk of revision surgery for dislocation at 1 year post primary index surgery with RoTHA compared to CoTHA with Odds Ratio of 0.3.Shaw et al. [35]20222247 patients (1724 CoTHA and 523 RoTHA), 3 surgeonsRetrospective cohort analysisTo compare pre-operative, post operative PROMs and complication rates between CoTHA and RoTHANo difference in PROMs (PROMIS-GH, PROMIS-MH, PROMIS-PH and HOOS, JR), reduced risk of dislocation


## 5. Conclusions

The evolution of surgical technology has resulted in the development of semiactive constrained robotic systems, and there is immense potential for this technology [38]. RoTHA should be therefore considered afresh, and there is good quality data suggesting superior and more accurate component positioning, restoration of the centre of rotation and native joint mechanics [39]. Recent evidence also suggests that RoTHA is associated with a significant reduction in the dislocation rate [34,35]. However, improvement in radiological outcomes has not been translated to improvement in PROMs or leg-length inequality correction. It needs to be acknowledged though that PROMs utilised in most studies have a substantial ceiling effect and are likely to have low discriminatory power, especially among high scores [40].

Therefore, results from prospective randomised controlled trials with longer-term follow up are needed to accurately evaluate and unmask any potential functional benefits of RoTHA. Furthermore, the role of registry data should also not be overlooked.

The cost effectiveness of RoTHA is an important topic, especially in relation to adopting this technology in publicly funded healthcare systems. There are significant costs associated with the equipment, staff, and training; however, stochastic models have shown this can be offset by a reduction in the length of stay and complications. As longer-term outcomes and complication rates emerge, the cost effectiveness will become clearer with further research into this area.

Finally, the enhanced planning and accuracy with robotic technology has brought to the fore the concept of personalised component positioning based on the patients’ phenotype. Overcoming the challenge of spinopelvic imbalance and impingement risk necessitates careful planning and high accuracy in executing the pre-operative plan. In this vein, robotic technology can offer a pragmatic way to tackle the challenges posed by abnormal motion in the spinopelvic kinetic chain. The integration of the virtual ROM tool has also enabled instant intra-operative feedback, based on which the arthroplasty surgeon can change the plan to avoid impingement and achieve the surgical targets.

## Figures and Tables

**Figure 1 medicina-58-01616-f001:**
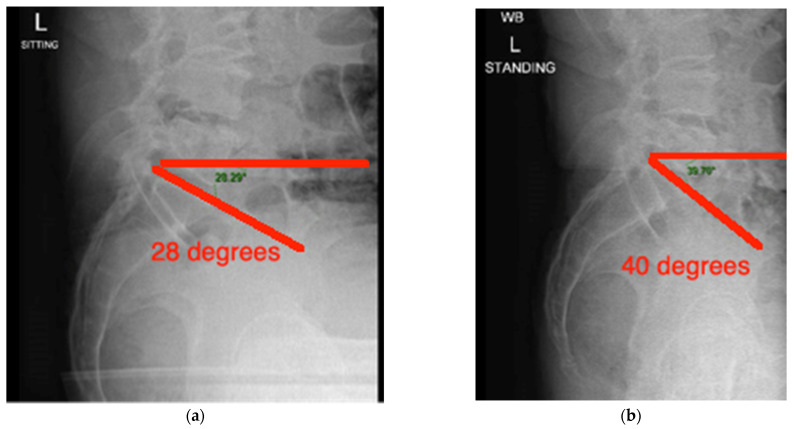
(**a**) Calculation of the sacral slope with sitting X-ray on the left (28 degrees) and (**b**) standing X-ray on the right (40 degrees). This signifies normal change from upright to sitting position as the difference is 12 (between 11 and 29).

**Figure 2 medicina-58-01616-f002:**
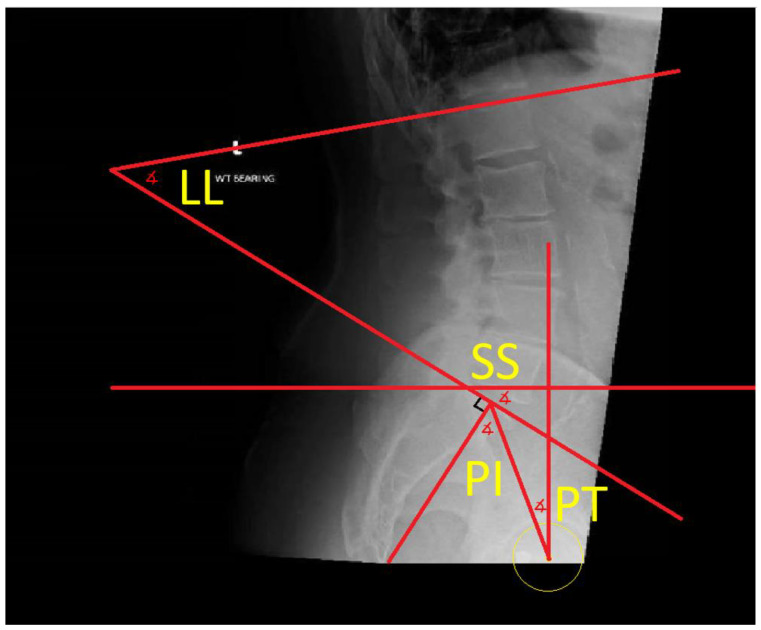
Annotated standing lateral lumbar spine xray defining the different spinopelvic parameters. Lumbar Lordosis (LL): the angle between superior end plates of L1 and S1; Sacral Slope (SS): the angle between a horizontal line and superior end plate of S1; Pelvic Tilt (PT): the angle between a vertical line and a line from femoral head (yellow circle) to midpoint of sacral endplate; Pelvic Incidence (PI): the angle between a line from the femoral head centre to midpoint of sacral endplate and a line perpendicular to the sacral end plate at its midpoint.

**Figure 3 medicina-58-01616-f003:**
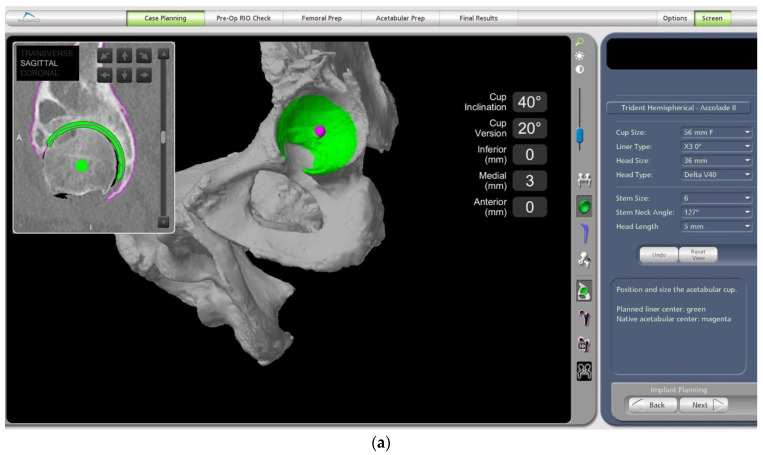
Evaluation of 3D bony coverage of the acetabular cup and careful planning in relation to restoring the native centre of rotation. (**a**) Sagittal. (**b**) Transverse. (**c**) Coronal.

**Figure 4 medicina-58-01616-f004:**
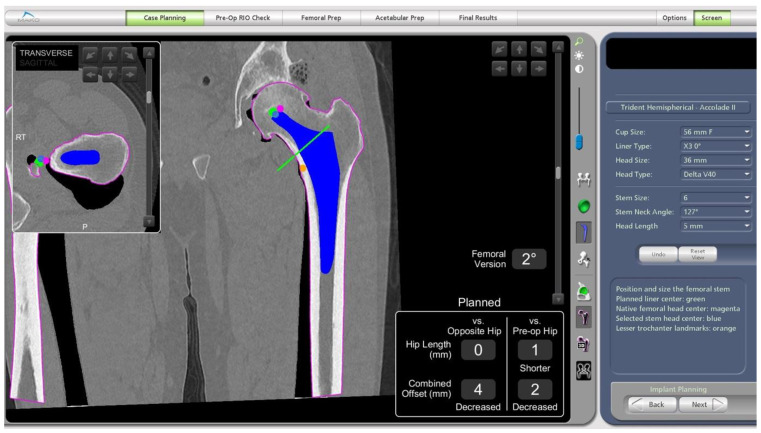
Preoperative screenshot showing planned femoral osteotomy with green line, femoral centre of rotation with green dot at tip of implant, planned leg length and combined offset changes at bottom right of screen and calculated femoral version just above.

**Figure 5 medicina-58-01616-f005:**
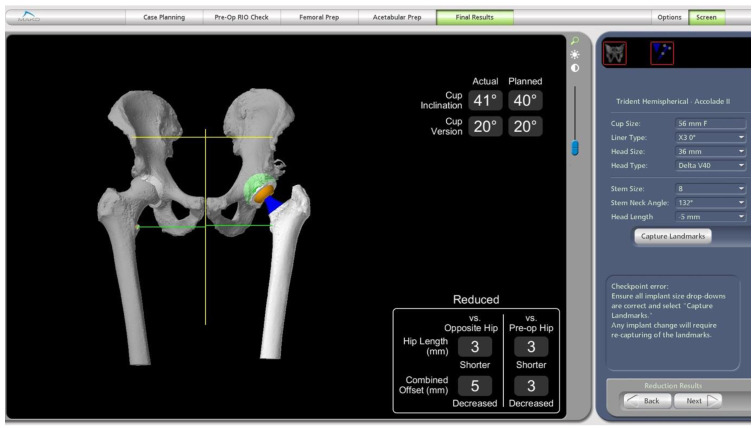
Intraoperative screenshot showing the actual cup position compared to the pre-operative plan. Leg length and combined offset changes are also displayed.

**Figure 6 medicina-58-01616-f006:**
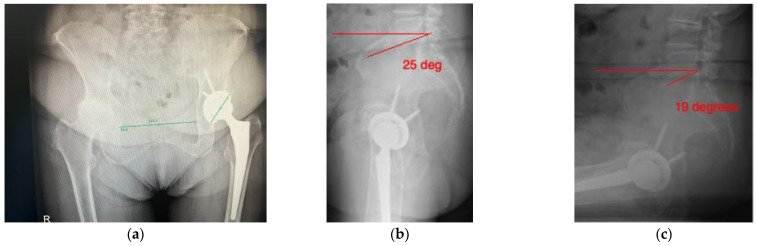
(**a**) AP pelvis radiograph; (**b**) standing lateral lumbar radiograph showing sacral slope of 25 degrees; (**c**) sitting lumbar radiograph showing sacral slope of 19 degrees.

**Figure 7 medicina-58-01616-f007:**
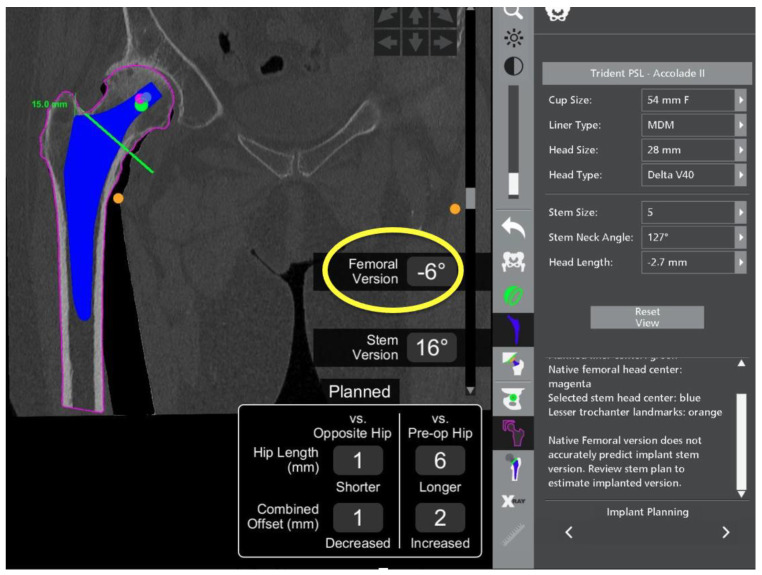
Native femoral version is −6 degrees (yellow circle).

**Figure 8 medicina-58-01616-f008:**
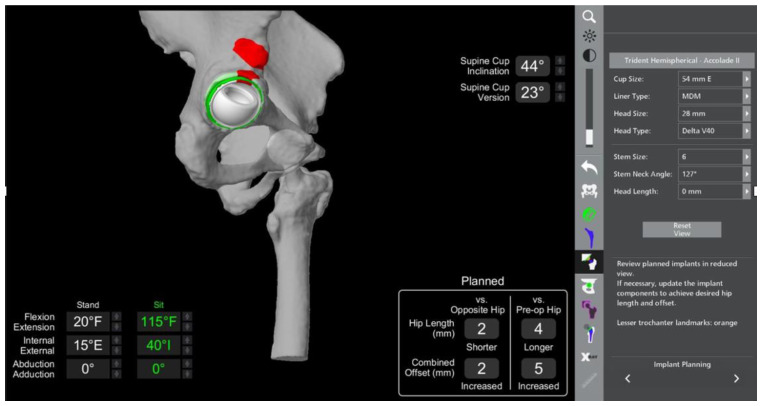
Bone-on-bone and implant-on-bone impingement in the postero-superior region of the acetabulum (in red) in deep flexion/external rotation.

**Figure 9 medicina-58-01616-f009:**
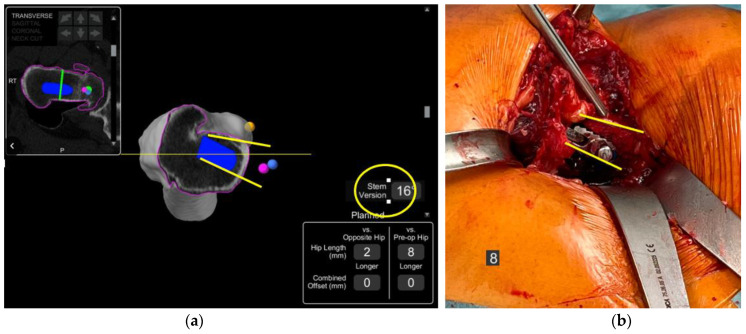
Correction of femoral retroversion from native −6 degrees to +16 degrees with the implant on the (**a**) robotic software (yellow circle) and (**b**) intraoperatively (yellow lines).

**Figure 10 medicina-58-01616-f010:**
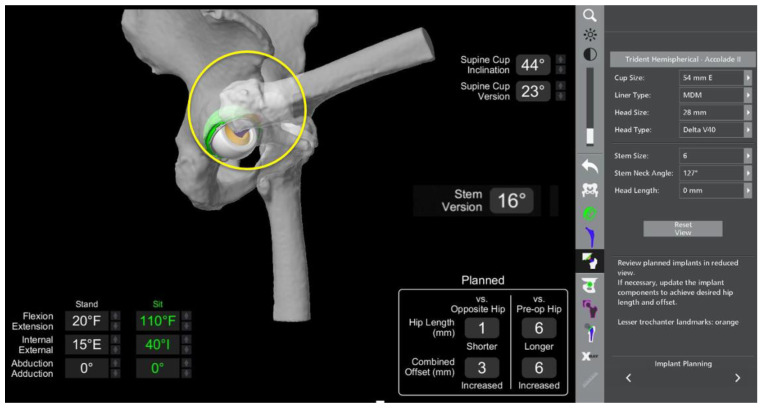
VROM in flexion with new femoral broach version of 16 degrees with no further evidence of impingement.

**Figure 11 medicina-58-01616-f011:**
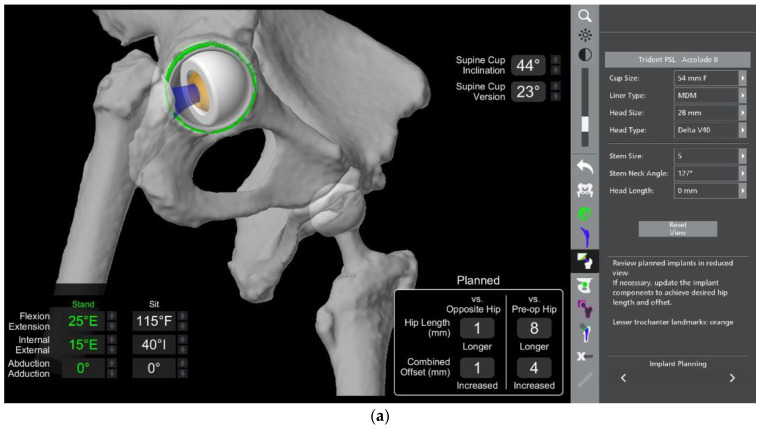
(**a**) VROM in extension with no obvious impingement. (**b**) VROM in extension (yellow circle) with subtraction of the implant showing a small area of anterior impingement (red area indicated by yellow arrow).

**Figure 12 medicina-58-01616-f012:**
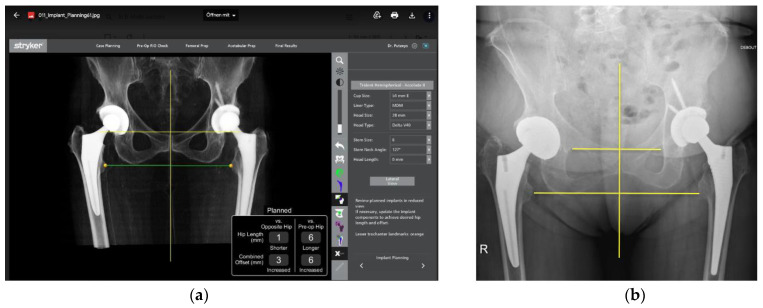
(**a**) Robotic software showing anticipated post-operative X-ray incorporating planned changes in leg length and offset. (**b**) Post-operative X-ray.

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
