# Peer review of "Robotic-Arm-Assisted Total Hip Arthroplasty: A Review of the Workflow, Outcomes and Its Role in Addressing the Challenge of Spinopelvic Imbalance"

_medicina, 2022, doi:10.3390/medicina58111616_

Round 1
Reviewer 1 Report
Thank you for the opportunity to review this article. The purpose of the narrative review is to summarize concepts on the use of robotics in hip replacement, particularly with regard to the usefulness of robotics in THA placement thought to spino-pelvic parameters.
In my opinion, the literature can benefit from a synthesis of evidence on this topic, and reading this review can provide a quick update and a starting point for more in-depth evaluations. Understanding the relationship between THA placement and sagittal balance of the spine is a challenge for hip surgeons, but could allow optimization of the prosthetic replacement process in the future, with fewer mechanical complications and better functional outcomes. Robotic navigation is the ideal weapon to use in this regard.
The article is well written and organized. The points covered in the discussion are relevant and interesting. However, I believe that:
1) more effort in describing the systems in use today (I am referring to "how it works") and that relevant figures of the preparation and use phases of the main systems are needed to promote understanding;
2) some illustrations are needed to more intuitively describe the spino-pelvic parameters, especially for the benefit of readers less familiar with the topic.
Thank you.
Reviewer 2 Report
Robotic-arm assisted Total Hip Arthroplasty: a review of the workflow, outcomes and its role in addressing the challenge of spinopelvic imbalance
Dear authors
It is a complete work, which presents valuable results for the scientific community. It has satisfactory results that meet the objectives of the study.
Here are some observations that could improve the content of the study. So that various technological areas understand the importance of your research.
• Line 35 to 36 - You mention some first uses and implementation of robots in surgery. Here you could put some references.
• Line 50- you mention MAKO Robotic-arm, is there any reference?
• In the introduction you could mention the importance of rehabilitation after an operation (THA). And you could mention the new soft robotic devices for these therapies. You can be guided by the following article:
“Pérez Vidal, A.F.; Rumbo Morales, J.Y.; Ortiz Torres, G.; Sorcia Vázquez, F.d.J.; Cruz Rojas, A.; Brizuela Mendoza, J.A.; Rodríguez Cerda, J.C. Soft Exoskeletons: Development, Requirements, and Challenges of the Last Decade. Actuators 2021, 10, 166. https://doi.org/10.3390/act10070166”.
• At the end of the introduction, you can also mention how many patients you analyzed for your study, as well as their age range.
The article can be accepted after having attended the observations
Round 2
Reviewer 1 Report
The authors have addressed all my concerns, thank you.